# DreamExplorations: Leveraging Suboptimal Noisy Trajectories in Offline RL

## Abstract

Exploration is a desirable characteristic in online reinforcement learning, where the online agent can interact with the environment, explore the diverse states, and update the policy. However, since the datasets of offline reinforcement learning are static and the traditional offline RL algorithms always rely on the relatively good quality of demo agents, it is very hard to explore the diversity of state space. In this paper, we have found out that in offline goal-conditioned reinforcement learning (OGCRL), we can theoretically leverage suboptimal/high noisy datasets for state exploration and we have designed a pipeline to use them. In this case, the highly noisy datasets which are always discarded and regarded as useless datasets in previous researches are used as exploration experts to keep improving the performances of offline reinforcement learning as we scale the sizes of suboptimal datasets. Experimental results demonstrate that our method consistently outperforms baselines and significantly improves models trained solely on high-quality data, especially in environments with large state spaces. This work highlights the untapped potential of imperfect data in enhancing the robustness and generalization of offline RL. We will open-source our code after publication.

## 1 Introduction

Reinforcement learning (RL) promises agents that can learn complex behaviors, but scaling RL beyond narrow settings remains a fundamental challenge. A key bottleneck is data: unlike language or vision, where large-scale datasets are abundant, RL typically relies on costly online interactions, sim-to-real transfer or carefully curated expert demonstrations. Offline goal-conditional reinforcement learning (OGCRL) Levine et al. (2020); Park et al. addresses this challenge by learning policies from previously collected data, conditioned on desired goals and using goal-conditional rewards. By decoupling policy learning from online interaction, OGCRL has emerged as one of the most promising directions for scalable RL in real worlds Ghosh et al. (2023); Park et al. (2023).

However, current OGCRL approaches face a critical limitation. When training under offline settings, a common practice is to rely on perfect trajectories—expert demonstrations or near-optimal rollouts that lead directly to the desired goals. While effective in narrow domains, this reliance often results in a restricted exploration space: policies overfit to a limited set of observed states and struggle to generalize when facing even minor deviations from the demonstrated trajectories. Current state-of-the-art OGCRL algorithms such as HIQL Park et al. (2023) and others depend critically on high-quality, near-optimal trajectories Park et al..

This assumption is problematic in practice. Real-world datasets—especially in robotics—rarely consist of perfect demonstrations. Instead, they are mixtures: some trajectories reflect expert-like performance, while many are noisy, inefficient, or outright failures Walke et al. (2023). Obtaining precise and optimal data is so difficult that suboptimal trajectories are often discarded entirely Yang et al. (2024); Hong et al. (2023); Beck. This wastes valuable information and severely limits scalability. We argue that diverse state exploration is the key to overcoming this limitation. Unlike online RL, where exploration can be actively performed in the environment, OGCRL must instead expand the effective coverage of the dataset. The suboptimal datasets collected by suboptimal policies can actually help explore more diverse states in the space, enabling better generalization. These "imperfect" demonstrations still encode essential knowledge on how to explore around the state space.

Figure 1: Different types of policies for OGCRL. We assume two distinct datasets: one from an optimal policy (short paths to the goal) and one from a suboptimal, noisy policy (near-random walks). While goal-conditional methods can use both, OGCRL performance typically degrades with the noisy data, leading to the common practice of discarding it. Our paper's core purpose is to show how to effectively utilize this suboptimal data rather than throwing it away.

Instead of treating them as noise and discarding them as traditional methods do Yang et al. (2024); Hong et al. (2023); Beck, we argue they can provide useful learning signals if integrated correctly.

This insight is inspired by the development of large-scale language models. The remarkable performance of GPT-4 Achiam et al. (2023) and DeepSeek-R1 Guo et al. (2025) stems not from carefully filtered, perfect corpora, but from massive and heterogeneous datasets. Traditionally, RL methods have addressed suboptimal datasets in two main ways: by discarding them when separable, and by making algorithms more robust when the data is inseparable. Instead of discarding these datasets, our approach aims to leverage and scale them to continuously enhance performance.

We introduce a method that injects controlled randomness into goal conditions and intermediate transitions. By perturbing trajectories with stochastic variations, the agent is encouraged to explore a wider state space beyond narrowly defined expert data, while still being guided by the original goal signals.The key idea is to let expert trajectories anchor the value landscape while allowing suboptimal ones to contribute to improve generalization and accelerate learning. Through theoretical analysis, we show that suboptimal trajectories can tighten value estimates and provide beneficial exploration pressure.

Our contributions are threefold:

1. **Theoretical Foundation**: We rigorously analyze the impact of suboptimal trajectories on OGCRL, establishing conditions under which they enhance policy learning.

2. **Algorithmic Innovation**: We introduce a novel OGCRL algorithm that incorporates structured randomness, effectively leveraging suboptimal datasets to improve performance.

3. **Empirical Validation**: We have proposed extensive experiments to thoroughly evaluate the effectiveness and robustness of our proposed algorithms.

By reframing suboptimal data as an asset, our work provides a principled and practical path toward scaling offline RL. Unlike most RL algorithms just trying to discard the suboptimal datasets, we argue that we can in fact learn from the them.

## 2 RELATED WORKS

Many robust algorithms have been proposed for mixed-quality learning in offline goal-conditioned reinforcement learning (OGCRL) when suboptimal datasets are inseparable. These include QRL Wang et al. (2023), CRL Eysenbach et al. (2022), IQL Kostrikov et al. (2021), GCBC Ding et al. (2019), and HIQL Park et al. (2023). Among these, HIQL Park et al. (2023) stands out as a state-of-the-art method across various OGCRL tasks. It leverages the value learning process of IQL

Kostrikov et al. (2021) and a hierarchical AWR Peng et al. (2019) policy structure to learn from goal-conditioned offline datasets. Although HIQL Park et al. (2023) is a highly robust method for mixed-quality learning, its performance still drops dramatically when trained with separable suboptimal datasets. In these cases, most prior works Yang et al. (2024); Hong et al. (2023); Beck focus on discarding or reducing the weight of suboptimal data. Unlike these methods, our approach aims to leverage suboptimal datasets to enhance algorithms that are already trained on near-perfect datasets. We believe that this is a critical scenario, especially in robotic applications where collecting near-perfect datasets is challenging Walke et al. (2023). Given that existing OGCRL algorithms focus primarily on handling mixed-quality datasets rather than using separable suboptimal data, our comparison with HIQL Park et al. (2023) and others serves as a meaningful benchmark against one of the strongest state-of-the-art methods in this domain.

## 3 PRELIMINARIES

### 3.1 PROBLEM SETTING

In the domain of **Offline Goal-Conditioned Reinforcement Learning (OGCRL)**, our primary objective is to learn an optimal policy from a precollected static dataset that can guide an agent to achieve specified goals. This problem is typically formalized by a Markov Decision Process $M = (S, A, \mu, p, r)$, where $S$ denotes the state space, $A$ denotes the action space, $\mu \in P(S)$ represents an initial state distribution, $p \in S \times A \to P(S)$ signifies the transition dynamics distribution, and $r(s, g)$ is a goal-conditioned reward function. The dataset $D$ consists of trajectories $\tau = (s_0, a_0, s_1, a_1, \ldots, s_T)$. We assume that the goal space $G$ is identical to the state space (i.e., $G = S$).

However, in practical applications, the quality of offline datasets often varies significantly, posing unique challenges for policy learning. Traditional offline reinforcement learning methods typically assume relatively consistent data quality or tend to discard samples deemed "low-quality" Yang et al. (2024); Hong et al. (2023); Beck. To fully leverage limited data resources, we face a critical question: How can we effectively utilize suboptimal datasets or even scale them?

Specifically, our policy consists of two distinct parts:

1. **Exploiting Robust Signals from Near-Expert Data**: Near-expert data are collected by a highly skilled agent, achieving a high success rate in the environment. Near-expert datasets are always strong and robust, we want to exploit the behavior pattern from them. However, they often cannot entirely cover the whole state space.

2. **Learning Generalizability by Stochastic Deviations**: Suboptimal data originates from an agent with very poor performance and an extremely low success rate. Although these data contain a large number of stochastic or even random deviations, they still include information regarding environmental dynamics and state space coverage. We want to explore the value functions of more state space and improve the generalization ability.

Our research aims to address how to avoid discarding these seemingly "unusable" low-quality data and instead ingeniously leverage them to assist and enhance the policy learning process. Our goal is to develop a novel method capable of learning an optimal goal-conditioned policy $\pi(a|s, g)$ from such a mixed dataset $D_{high} \cup D_{low}$. By effectively utilizing the latent information contained within the low-quality data, we expect to significantly improve the success rate of the final policy, especially in scenarios where high-quality data is scarce or costly to acquire.

### 3.2 HIERARCHICAL IMPLICIT Q-LEARNING(HIQL)

To effectively address the challenges of learning robust policies from mixed quality offline datasets in goal-conditioned reinforcement learning, we first explore **Hierarchical Implicit Q-Learning (HIQL Park et al. (2023))**. HIQL Park et al. (2023) is recognized as **one of the State-of-the-Art (SOTA) algorithms** in Offline Goal-Conditioned Reinforcement Learning (OGCRL). However, despite its strong performance in many aspects, HIQL Park et al. (2023) can be sensitive to the quality of the datasets, and it cannot leverage the suboptimal datasets to improve the performances. This characteristic poses a significant challenge for its application in mixed-quality real-world datasets.

### 3.2.1 Foundation: Action-Free Implicit Q-Learning (IQL)

At its core, HIQL Park et al. (2023) is based on the action-free variant of Implicit Q-Learning (IQL) Kostrikov et al. (2021), which is a famous offline RL algorithm. The value function of both HIQL and IQL is optimized using the following expectile regression loss:

$$\mathcal{L}_V(\theta_V) = \mathbb{E}_{(s,s')\sim\mathcal{D}_S, g\sim p(g|\tau)} \left[ L_\tau^2(r(s,g) + \gamma \bar{V}_{\theta_V}(s',g) - V_{\theta_V}(s,g)) \right], \tag{1}$$

where $L_\tau^2(x) = |\tau - \mathbf{1}(x < 0)|x^2$ is the expectile loss with parameter $\tau \in [0.5, 1)$, and $\bar{V}_{\theta_V}$ denotes the target value network. This objective directly utilizes next-state values for backups, bypassing the need for action information during value function fitting.

### 3.2.2 Hierarchical Policy Extraction

HIQL Park et al. (2023) decomposes the overall policy of IQL Kostrikov et al. (2021) into two levels: a high-level policy $\pi_h$ responsible for generating representations of intermediate sub-goals, and a low-level policy $\pi_\ell$ tasked with executing primitive actions to reach these sub-goals. Both policies are extracted from the learned goal-conditioned value function $V_{\theta_V}(s,g)$ using Advantage-Weighted Regression (AWR) style objectives.

The high-level policy $\pi_h(s_{t+k}|s_t, g)$ is trained to predict optimal $k$-step sub-goals $s_{t+k}$ towards the ultimate goal $g$, with the following optimization objective:

$$J_{\pi_h}(\theta_h) = \mathbb{E}_{(s_t, s_{t+k}, g)} \left[ \exp(\beta \cdot \tilde{A}_h(s_t, s_{t+k}, g)) \log \pi_{h,\theta_h}(s_{t+k}|s_t, g) \right], \tag{2}$$

where $\tilde{A}_h(s_t, s_{t+k}, g) \approx V_{\theta_V}(s_{t+k}, g) - V_{\theta_V}(s_t, g)$ serves as the advantage function.

The low-level policy $\pi_\ell(a_t|s_t, s_{t+k})$ is trained to produce primitive actions $a_t$ to reach the immediate sub-goal $s_{t+k}$, with the objective:

$$J_{\pi_\ell}(\theta_\ell) = \mathbb{E}_{(s_t, a_t, s_{t+1}, s_{t+k})} \left[ \exp(\beta \cdot \tilde{A}_\ell(s_t, a_t, s_{t+k})) \log \pi_{\ell,\theta_\ell}(a_t|s_t, s_{t+k}) \right], \tag{3}$$

where $\tilde{A}_\ell(s_t, a_t, s_{t+k}) \approx V_{\theta_V}(s_{t+1}, s_{t+k}) - V_{\theta_V}(s_t, s_{t+k})$ is its corresponding advantage. Here, $\beta \in \mathbb{R}_{\geq 0}$ is an inverse temperature parameter.

## 4 Leveraging Suboptimal Datasets for OGCRL

### 4.1 Motivation

In offline reinforcement learning, high-quality datasets ($D_{high}$) often lack state-space coverage, leading to unreliable value estimates for unvisited states. In contrast, our low-quality dataset ($D_{low}$), despite its highly suboptimal actions, inherently explores a broader range of states. We posit that $D_{low}$ can provide crucial information to improve value function estimates for states underrepresented in $D_{high}$, thus mitigating the uncertainty in $V(s)$ in a wider distribution. Although existing methods typically discard such suboptimal data, our core motivation is to leverage these "unusable" samples to enhance policy generalization, especially when comprehensive high-quality data are scarce.

### 4.2 Didactic Example

To investigate this, we conducted a 2D simulation that models goal-conditioned rewards, varying action policy noise (Fig. 2). The expert values are learned by trajectories heading towards the goal with a noise of 1.0, the suboptimal is with a noise of 10.0 and the combined value functions will predict the values based on the frequency of the trajectories (weighted estimation of both based on the visit counts). The left is the visualization of value functions and the green points are the goals. The "Ground Truth" is the ground truth value function. The "Expert Dataset" is the value functions learned from near-expert datasets, the "Suboptimal Dataset" is from suboptimal datasets and "Combined Value Function" is the weighted estimation of both based on the visit counts. The medium results demonstrate that combining two value functions reduces the MSE between our estimates and ground truth values (the combined estimation has the lowest MSE). The right figure shows the difference of visit counts of certain states(red = expert visited more, blue = suboptimal visited more). We

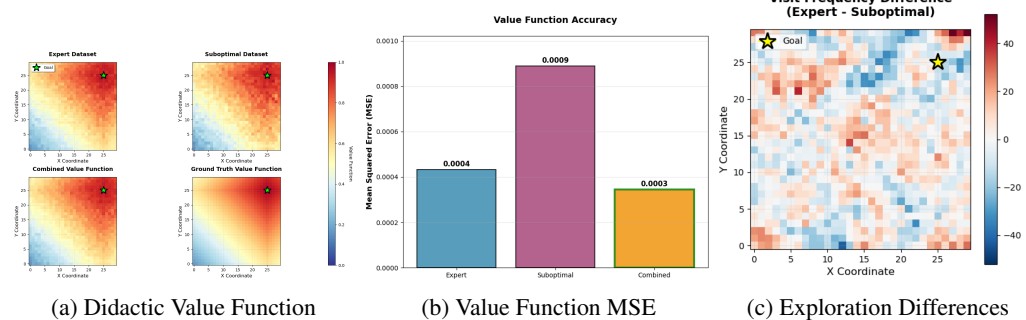

(a) Didactic Value Function      (b) Value Function MSE      (c) Exploration Differences

Figure 2: The left figure are the visualization of value functions and the right two figures are the statistical results of the estimations of the value functions and the exploration differences. In this picture we want to give an instinct that different types of datasets may have complementary strengths.

can see that expert datasets and suboptimal datasets may have complementary strengths/confidence in different state spaces. In conclusion, in the OGCRL setting, suboptimal datasets can empower value functions with diversity in state estimation, resulting in higher accuracy.

### 4.3 THEORETICAL ANALYSIS

When we learn a normal value function Sutton et al. (1998), we can estimate the learned value function $\hat{V}$ as the Eq. 4. The $V^\star$ is the ground truth of the value function, the $b_{action}$ comes from the bias from the stochasticity of actions, the $b_{bp}$ comes from the bias from bootstrapping, the $b_{en}$ comes from the stochasticity of the environment and the $\varepsilon$ comes from the noise.

$$\hat{V} = V^\star + b_{action} + b_{bp} + b_{en} + \varepsilon \tag{4}$$

When we learn an action-free value function in OGCRL as Eq. 1, the $b_{action} = 0$ Park et al. (2023); Ghosh et al. (2023) (we no longer have action terms in the value functions). According to Bickel & Freedman (1981); Singh (1981), the $b_{bp}$ converges at the rate $O\left(\frac{1}{\sqrt{n}}\right)$. In deterministic environment, the $b_{en} = 0$ and the $b_{en}$ will often optimistically biased in a stochastic environment. However, the authors in Ghosh et al. (2023) have used experiments to show that $b_{en}$ in action-free value learning (Eq.1) are often nuances ($\approx 0$) even in stochastic environments. In other words, $b_{en} \approx 0$ in all both deterministic and stochastic environments according to Ghosh et al. (2023). When noise $\varepsilon$ is a normal distribution, we can always approximate that noise converges at the rate $O\left(\frac{1}{\sqrt{n}}\right)$. We can also infer from the discussions above that the accuracy of value functions is mostly related to the distributions of datasets. As a result, the value functions learned from suboptimal datasets are useful and can compensate for the estimations of states that the near-expert trajectories have rarely covered.

### 4.4 MERGED ESTIMATION IN OGCRL

After the previous analysis, we have already understood the compositions of $\hat{V}$. Fix a particular state–goal pair $(s, g)$ and denote the true goal-conditioned value by $V^\star(s, g)$. Let $\hat{V}_{high}(s, g)$ and $\hat{V}_{low}(s, g)$ be value estimates produced by fitting the same value-regression objective (see Eq. 1) on $\mathcal{D}_{high}$ and $\mathcal{D}_{low}$, respectively. For brevity we drop the explicit $(s, g)$ when context is clear and write

$$\hat{V}_i = V^\star + b + \varepsilon \tag{5}$$

where $b$ is the bias ($b \approx b_{bp}$) and $\varepsilon$ is a zero-mean random error. If we want to leverage both the near-expert and suboptimal datasets, we may want to learn a mixed value function from both of the datasets. We consider that $V_{high}$ is the theoretical values from high quality (near-expert) datasets and $V_{low}$ from low quality (suboptimal) datasets and the $w$ is best weight of $V_{high}$ and $1 - w$ is best weight of $V_{low}$ for each $(s, g)$ pair. And we have Eq. 6.

$$\hat{V}_w \triangleq w\,\hat{V}_{high} + (1-w)\,\hat{V}_{low}, \qquad w \in [0, 1]. \tag{6}$$

The mean squared error of $\hat{V}_w$ decomposes as

$$\mathrm{MSE}(w) = \left(wb_{high} + (1-w)b_{low}\right)^2 + w^2\varepsilon_{high}^2 + (1-w)^2\varepsilon_{low}^2. \tag{7}$$

## 4.5 OPTIMAL LINEAR MIXING AND INTERPRETATION

Given the approximations for the bias and variance terms based on the sizes of the dataset $n_1$ (high-quality dataset) and $n_2$ (low-quality dataset), we have:

$$b_{\text{high}} = \frac{c_1}{\sqrt{n_1}}, \quad b_{\text{low}} = \frac{c_2}{\sqrt{n_2}} \tag{8}$$

$$\varepsilon^2_{\text{high}} = \frac{d_1}{n_1}, \quad \varepsilon^2_{\text{low}} = \frac{d_2}{n_2} \tag{9}$$

where $c_1, c_2, d_1, d_2$ are environment related constants (In the same environment, $c_1 = c_2, d_1 = d_2$).

**Lemma 1 (optimal weight).** Minimizing equation 7 in $w$ (over $\mathbb{R}$) produces the closed form

$$w^* = \frac{\frac{c_1 c_2}{\sqrt{n_1 n_2}} - \frac{c_2^2}{n_2} + \frac{d_2}{n_2}}{\frac{c_1^2}{n_1} + \frac{c_2^2}{n_2} + \frac{d_1}{n_1} + \frac{d_2}{n_2}} \tag{10}$$

Let $p = \frac{n_1}{n_2}$ be the ratio of the sizes of the dataset. We can rewrite the optimal weight in terms of $p$, we have $\frac{n_2}{n_1} = \frac{1}{p}$. Substituting:

$$w^* = \frac{c_1 c_2 \frac{1}{\sqrt{p}} - c_2^2 + d_2}{\frac{c_1^2}{p} + c_2^2 + \frac{d_1}{p} + d_2} \tag{11}$$

**Interpretation.** We can see in the expression Eq. 11 that the more times high quality datasets have sampled the particular state (the higher $p$ is), the bigger $w^\star$ should be (In the same environment, $c_1 \approx c_2$ and $d_1 \approx d_2$). Details in the Appendix.

## 4.6 FROM ANALYTIC WEIGHT TO THE NOVELTY-RATIO NETWORK

We operationalize the mixing by parameterizing the value used for policy extraction as in the equation below($(1 - R_\psi(s))$ for $w^*$, $R_\psi(s)$ is a network with a scalar output predicting a float weight):

$$V_{\text{mix}}(s, g) = (1 - R_\psi(s)) V_{\text{Exploit}}(s, g) + R_\psi(s) V_{\text{Explore}}(s, g), \tag{12}$$

**Interpretation.** We want to learn a network $R_\psi$ from the task that the more often a particular state is sampled in high quality datasets than in low quality datasets, the $(1 - R_\psi(s))$ should be greater.

## 5 OUR METHODS

Our method is based on the HIQL Park et al. (2023) method. However, we have adapted it for suboptimal datasets. We aim to further improve the performance of offline goal-conditional reinforcement learning policies using the suboptimal datasets.

### 5.1 DECOUPLED VALUE LEARNING

We train two separate value functions using the same value regression objective but with different datasets. Given a state–goal pair $(s, g)$ and its successor state $s'$, the value loss is defined as Eq. 1. To model exploitation-oriented and exploration-oriented value estimates, we have the following.

- For **Value$_{\text{Exploit}}$**, $(s, g)$ is sampled from the high-quality dataset $\mathcal{D}_{\text{high}}$:

$$\mathcal{L}_V^{\text{high}}(\theta_V) = \mathbb{E}_{(s,s') \sim \mathcal{D}_{\text{high}}, \, g \sim p(g|\tau)} \left[ L_\tau^2 \big( r(s, g) + \gamma \bar{V}_{\theta_V}(s', g) - V_{\theta_V}(s, g) \big) \right]. \tag{13}$$

- For **Value$_{\text{Explore}}$**, $(s, g)$ is sampled from the low-quality dataset $\mathcal{D}_{\text{low}}$:

$$\mathcal{L}_V^{\text{low}}(\theta_V) = \mathbb{E}_{(s,s') \sim \mathcal{D}_{\text{low}}, \, g \sim p(g|\tau)} \left[ L_\tau^2 \big( r(s, g) + \gamma \bar{V}_{\theta_V}(s', g) - V_{\theta_V}(s, g) \big) \right]. \tag{14}$$

The resulting Value$_{\text{Exploit}}$ captures the expected returns under high-quality behavior policies, while Value$_{\text{Explore}}$ reflects the potential returns in more exploratory behaviors.

## 5.2 Novelty Ratio

We additionally train a novelty estimation network $R_\psi : \mathcal{S} \to [0, 1]$ that maps a state $s$ to a scalar novelty ratio (similar to the ideas in didactic examples that we trust the value functions of the most visited trajectories). The network is supervised using states sampled from both the high-quality dataset $\mathcal{D}_{\text{high}}$ and the low-quality dataset $\mathcal{D}_{\text{low}}$. The loss of novelty ratio is defined as

$$\mathcal{L}_{\text{novelty}}(\psi) = \mathbb{E}_{s \sim \mathcal{D}_{\text{high}}} \left[ \left( R_\psi(s) - A^- \right)^2 \right] + \mathbb{E}_{s \sim \mathcal{D}_{\text{low}}} \left[ \left( R_\psi(s) - A^+ \right)^2 \right], \tag{15}$$

where the target value is $A^-$ for high-quality states and $A^+$ for low-quality states ($0 \leq A^- < A^+ \leq 1$, where $A^-$ and $A^+$ are adjustable hyperparameters). We can think of $R_\psi(s)$ as the novelty ratio, the less possible agents have already learned about the states in the perfect datasets, the more agents should rely on the suboptimal datasets, and the bigger $R_\psi(s)$. In some cases, $A^-$ can be simply 0 and $A^+$ can be simply 1 or 0.5. This encourages $R_\psi$ to produce lower novelty scores for states of $\mathcal{D}_{\text{high}}$ and higher novelty scores for states of $\mathcal{D}_{\text{low}}$.

## 5.3 Novelty-Value-Guided Policy Learning

In our proposed method, we retain the same high-level policy structure $\pi_h$ as in HIQL Park et al. (2023), as it depends on fixed subgoal-related hyperparameters. Specifically, $\pi_h(s_{t+k}|s_t, g)$ is still trained by the the objective in Eq. 2, using the advantages computed by the goal-conditioned value function.

The modification lies in the low-level policy $\pi_\ell(a_t|s_t, s_{t+k})$. During training, the state $s_t$ is always sampled from the high-quality dataset $\mathcal{D}_{\text{high}}$, ensuring that primitive actions are learned in reliable state distributions. However, the advantage computation is no longer based on a single value function $V_{\theta_V}$. Instead, we use a novelty ratio-weighted mixture of the exploitation and exploration value functions:

$$V_{\text{mix}}(s, g) = \left( 1 - R_\psi(s) \right) \cdot V_{\text{Exploit}}(s, g) + R_\psi(s) \cdot V_{\text{Explore}}(s, g), \tag{16}$$

where $R_\psi(s) \in [0, 1]$ is the novelty ratio predicted by the network introduced in Eq. 15.

Accordingly, the low-level policy objective becomes:

$$J_{\pi_\ell}(\theta_\ell) = \mathbb{E}_{(s_t, a_t, s_{t+1}, s_{t+k}) \sim \mathcal{D}_{\text{high}}} \left[ \exp \left( \beta \cdot \tilde{A}_\ell(s_t, a_t, s_{t+k}) \right) \log \pi_{\ell, \theta_\ell}(a_t|s_t, s_{t+k}) \right], \tag{17}$$

where the advantage is now defined as:

$$\tilde{A}_\ell(s_t, a_t, s_{t+k}) \approx V_{\text{mix}}(s_{t+1}, s_{t+k}) - V_{\text{mix}}(s_t, s_{t+k}). \tag{18}$$

This design allows the low-level policy to leverage high-quality state action supervision while still incorporating exploration signals in proportion to the predicted novelty of each state.

However, because the novelty ratio is unstable in the beginning epochs. So, we don't introduce any kind of suboptimal datasets to policy learning procedures and rigorously follow the original policy learning functions Eq.2 and Eq.3 until all of the near perfect datasets have been learned once.

## 6 Experiments

In order to further prove the effectiveness of our methods and the rules behind the performances under suboptimal datasets, we conduct experiments to evaluate our approach. To begin, we generate both suboptimal and near-perfect datasets following the dataset generation pipeline in OGbenchPark et al.. The HIQL algorithm Park et al. (2023) in OGbenchPark et al. has added a noise of "0.2" to demo agents in locomotion tasks to generate near-perfect policies. We have added a noise of "2.0"(point-maze "30") to the demo agents to generate the suboptimal policies. Our experiments are based on locomotion tasks in OGbenchPark et al. because OGbenchPark et al. is the SOTA benchmark in OGCRL and locomotion tasks can be easier to analyze in terms of the sizes of states and difficulties of tasks. Because the performances of offline reinforcement learning are unstable, we have trained eight times per task (the same as the benchmark) and computed the means and variances of each performance of the tasks.

## 6.1 OVERALL BASELINE COMPARISON

We conducted experiments using the tasks containing 'explore' (random noise policy) datasets in the OGBench Park et al. plus some others. The details and reasons of task selections and datasets generations will be shown in the Appendix. Except re-running results for HIQLPark et al. (2023) and our methods, we take the results of the method from the benchmarkPark et al. paper. We have made comparisons to robust methods including GCBC Lynch et al. (2020); Ghosh et al. (2019), GCIVL Kostrikov et al. (2021); Park et al. (2023), GCIQL Kostrikov et al. (2021); Park et al. (2023), QRL Wang et al. (2023), CRL Eysenbach et al. (2022). We show that, by using some trajectories collected by suboptimal policy, we can reach the best results among all the algorithms.

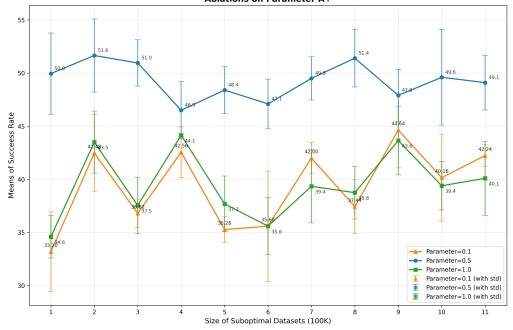

Table 1: Comparison Results.

| Task | GCBC | GCIVL | GCIQL | QRL | CRL | HIQL | Ours |
|------|------|-------|-------|-----|-----|------|------|
| human-medium | 8 | 24 | 27 | 21 | 60 | 76 | 80 |
| human-large | 1 | 2 | 2 | 5 | 24 | 20 | 27 |
| human-giant | 0 | 0 | 0 | 1 | 3 | 10 | 17 |
| point-teleport | 31 | 44 | 25 | 9 | 4 | 20 | 32 |
| ant-medium | 29 | 72 | 71 | 88 | 95 | 96 | 96 |
| ant-giant | 24 | 16 | 34 | 75 | 83 | 69 | 74 |
| ant-large | 0 | 0 | 0 | 14 | 16 | 82 | 92 |
| ant-teleport | 26 | 39 | 35 | 35 | 53 | 47 | 52 |

Figure 3: Ablation Study on A+

Table 2: Warmup Ablations on 'Ant-Teleport' task

| Suboptimal Samples (in 100k) | 1 | 2 | 3 | 4 | 5 | 6 | 7 | 8 | 9 | 10 |
|------|------|------|------|------|------|------|------|------|------|------|
| With Warmup (success rate %) | 50.95 | 46.50 | 48.40 | 47.10 | 49.50 | 51.40 | 47.90 | 49.60 | 49.10 | 50.20 |
| Without Warmup (success rate %) | 41.10 | 36.15 | 34.75 | 36.30 | 35.40 | 36.55 | 36.20 | 38.55 | 33.85 | 33.35 |

## 6.2 ABLATION STUDY

**Ablation Study on Decoupled Value Structure.** We conduct comparison experiments in Table. 3 with HIQLPark et al. (2023) baseline (we have removed the decoupled value functions and only learned with the original value functions). The baseline was computed using the mixture of near-optimal(1 million pairs) and suboptimal datasets(1 million pairs) and. In this experiment, we have used the best suboptimal datasets - optimal datasets ratio to compute the results of our methods. We can see in Table. 3 that we have outperformed the baselines(HIQL) in all tasks.

**Ablation Study on All Expert Data.**We have also performed experiments to compare our methods with only using the near-perfect datasets and discarding the suboptimal datasets collected by noisy demo policy (The ablations column in Table. 3). We can also see in the Table. 3 that we have outperformed them in 7 out of 8 tasks (the remaining one is very close).

**Ablation Study on A+.** An ablation study on the parameter $A+$ was conducted on the 'Ant-Teleport' task. As shown in Fig. 3, we can see that as $A+$ increases from 0.1 to 1.0, the success rate first increases and then decreases. This is because $A+$ is closely related to the newly explored states of suboptimal trajectories. $A+$ influences the extent to which the algorithm learns from suboptimal datasets, thus affecting the mean success rate. However, it does not significantly change the variance. We found that setting $A+$ to 0.5 is a robust choice, while $A-$ is always set to 0.

**Ablation Study on Warmup.** In our algorithm, we have first used all expert datasets to warm up the value functions and then use the decoupled value strategy. In order to prove the effectiveness, we have also done experiments shown in Table. 2. We have computed the mean success rate based on 8 run times. As we can see, warmup is a necessary stategy in the whole pipeline.

## 6.3 STATE SPACE VS. PERFORMANCE

In Section 4 (Didactic Examples), we have analyzed that the increase of performance comes from the information of the new states brought by suboptimal datasets. If the state space is larger, more new states will be possibly explored by the suboptimal datasets. So, we can also infer that, compared to only using near-perfect datasets, the larger the state-space of the task is, the more will our algorithm

Table 3: Ablation Results 2

| Task | Baseline (HIQL) | Ablations (All Expert data) | Ours Methods |
|---|---|---|---|
| human-medium | 34.80 | 75.55 | 79.99 |
| human-large | 14.85 | 19.90 | 26.75 |
| human-giant | 2.70 | 9.70 | 17.30 |
| point-teleport | 22.80 | 20.00 | 32.00 |
| ant-medium | 89.35 | 95.99 | 95.80 |
| ant-giant | 24.65 | 68.55 | 74.35 |
| ant-large | 81.35 | 82.30 | 91.75 |
| ant-teleport | 48.80 | 47.15 | 51.65 |

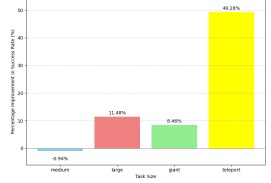

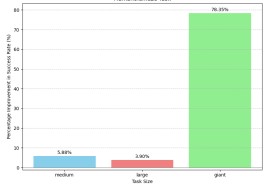

Figure 4: Ant-Maze.          Figure 5: Humanoid.

increases the performance. We can see Fig. 4, 5 for the performance improvements. The Y-axis shows the increase of performances over only using near-perfect datasets. As the X-axis value increases, the state space becomes larger for each task (Antmaze & Humanoidmaze). We can see that there is a overall improvement as X-axis gets bigger.

## 6.4 SCALING SUBOPTIMAL DATASET

In this experiment, we want to check whether scaling the suboptimal dataset can help to keep improving the performance of the offline GCRL in Fig. 6. We can see that the success rates of large state space tasks (c,e) can continue increasing, while the success rates of relatively small state space tasks (a,b) will reach an optimal success rate and then decline. This is due to the limited state space, as said in theoretical analysis and didactic examples. When the unexplored states become less, the information that can be gained from suboptimal datasets will also become less. An important question arises: Why do success rates in small state space decline after reaching a certain point in some tasks? This phenomenon occurs because neural networks cannot perfectly approximate arbitrary functions. As a result, the value network tends to place higher weights on losses from suboptimal datasets while reducing the weights on losses from near-perfect datasets. When more suboptimal datasets are incorporated, the value functions associated with actions in near-perfect datasets may suffer larger losses, ultimately leading to decreased performance.

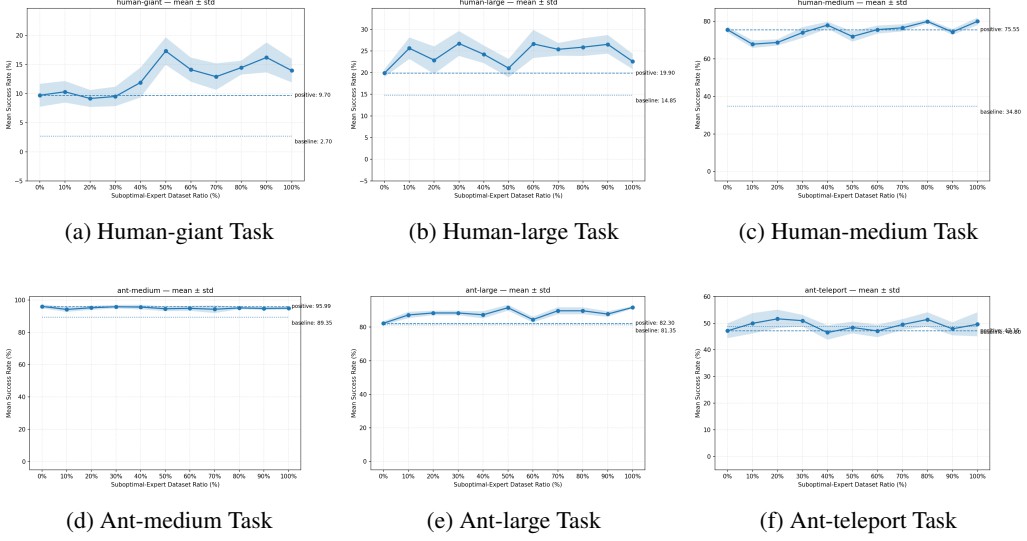

(a) Human-giant Task          (b) Human-large Task          (c) Human-medium Task

(d) Ant-medium Task          (e) Ant-large Task          (f) Ant-teleport Task

Figure 6: The X-axis is the ratio of suboptimal datasets to expert datasets (pairs) and the Y-axis is the mean success rate and standard deviation of 8 runs for each task.

## 7 CONCLUSION

We have proposed a method to take advantage of the trajectories collected from suboptimal policy to further improve the performances of the SOTA method in OGCRL. While most of existing methods try to use as perfect datasets as possible, we introduce a way to leverage the suboptimal datasets and possibly scaling them instead of discarding them in traditional pipelines. We have presented a concrete theoretical analysis, didactic examples, and experiments to prove the effectiveness of our methods. Results have shown our potential in scaling the suboptimal datasets in robot learning.

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

# A APPENDIX

## A.1 HOW CAN WE LEVERAGE SUBOPTIMAL DATA?

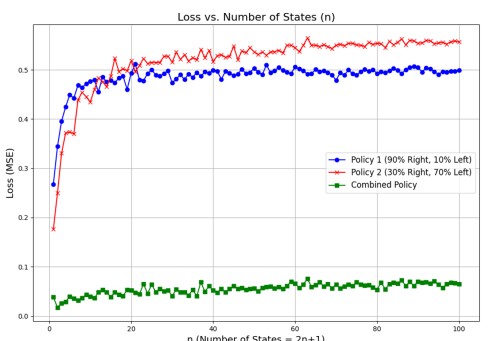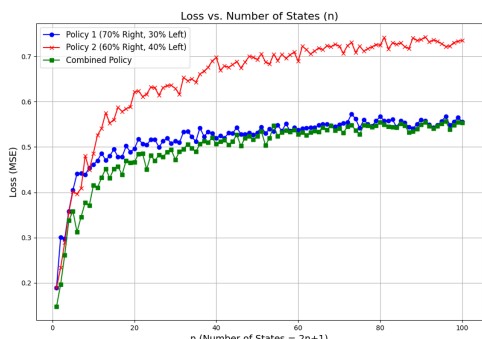

Figure 7: Didactic example for our proposed structure.

We have set a didactic simulation example to simulate the estimation of the value functions (results in Fig. 7). We analyze an agent exploring a 1D state space of $2n + 1$ states, starting from the center (state 0). Each state has a constant ground truth value $V$. The agent performs N independent random walks, each walk lasting up to $2n$ steps or until an endpoint is reached. When a state is visited, the agent observes a noisy value ($V + \mathcal{N}(0, \sigma^2)$); unvisited states are assigned a random noise value. The final estimate for each state is the average of all observations. We compare different policies, which are random walks with different right/left probabilities ($p_1, 1 - p_1$ and $p_2, 1 - p_2$), and a Combined Policy. The Combined Policy takes the final estimate for each state from the individual policy that visited it more frequently. Our goal is to analyze and prove the relationship between the Mean Squared Error (MSE) of these three policies. As shown in Fig. 7, our combined policy always has a smaller loss than the others, meaning that when we use the suboptimal datasets(lower probabilities to go right), we can make our final value function estimations better than only using the optimal/better datasets.

## A.2 OPTIMAL WEIGHT DERIVATION

Given the approximations for the bias and variance terms based on the sizes of the dataset $n_1$ (high-quality dataset) and $n_2$ (low-quality dataset), we have:

$$b_{\text{high}} = \frac{c_1}{\sqrt{n_1}}, \quad b_{\text{low}} = \frac{c_2}{\sqrt{n_2}} \tag{19}$$

$$\varepsilon_{\text{high}}^2 = \frac{d_1}{n_1}, \quad \varepsilon_{\text{low}}^2 = \frac{d_2}{n_2} \tag{20}$$

where $c_1, c_2, d_1, d_2$ are constants (In the same environment,$c_1 \approx c_2, d_1 \approx d_2$)..

Substituting these into the MSE expression:

$$\text{MSE}(w) = \left( w \frac{c_1}{\sqrt{n_1}} + (1 - w) \frac{c_2}{\sqrt{n_2}} \right)^2 + w^2 \frac{d_1}{n_1} + (1 - w)^2 \frac{d_2}{n_2} \tag{21}$$

Taking the derivative with respect to $w$ and setting it to zero:

$$\frac{d}{dw} \text{MSE}(w) = 2 \left( w \frac{c_1}{\sqrt{n_1}} + (1 - w) \frac{c_2}{\sqrt{n_2}} \right) \cdot \left( \frac{c_1}{\sqrt{n_1}} - \frac{c_2}{\sqrt{n_2}} \right) + 2w \frac{d_1}{n_1} - 2(1 - w) \frac{d_2}{n_2} = 0 \tag{22}$$

Expanding and simplifying:

$$w \frac{c_1^2}{n_1} + (1 - w) \frac{c_1 c_2}{\sqrt{n_1 n_2}} - w \frac{c_1 c_2}{\sqrt{n_1 n_2}} - (1 - w) \frac{c_2^2}{n_2} + w \frac{d_1}{n_1} - (1 - w) \frac{d_2}{n_2} = 0 \tag{23}$$

Collecting terms with $w$:

$$w \left[ \frac{c_1^2}{n_1} + \frac{c_2^2}{n_2} + \frac{d_1}{n_1} + \frac{d_2}{n_2} \right] = \frac{c_1 c_2}{\sqrt{n_1 n_2}} - \frac{c_2^2}{n_2} + \frac{d_2}{n_2} \tag{24}$$

Therefore, the optimal weight is:

$$w^* = \frac{\frac{c_1 c_2}{\sqrt{n_1 n_2}} - \frac{c_2^2}{n_2} + \frac{d_2}{n_2}}{\frac{c_1^2}{n_1} + \frac{c_2^2}{n_2} + \frac{d_1}{n_1} + \frac{d_2}{n_2}} \tag{25}$$

This can also be expressed in terms of the original bias and variance terms:

$$w^* = \frac{b_{\text{high}} b_{\text{low}} - b_{\text{low}}^2 + \varepsilon_{\text{low}}^2}{b_{\text{high}}^2 + b_{\text{low}}^2 + \varepsilon_{\text{high}}^2 + \varepsilon_{\text{low}}^2} \tag{26}$$

### A.3 PSEUDOCODE

In order to make the algorithms clearer, we have written down the pseudocode in the Appendix.

---

**Algorithm 1:** OGCRL Using Suboptimal Datasets

---

**Input:** High-quality dataset $\mathcal{D}_{\text{high}}$, low-quality dataset $\mathcal{D}_{\text{low}}$, discount factor $\gamma$, inverse temperature $\beta$

**Output:** Trained policies $\pi_h$, $\pi_\ell$

1 **Initialize:** value functions $V_{\text{Exploit}}$, $V_{\text{Explore}}$, novelty network $R_\psi$, policies $\pi_h$, $\pi_\ell$;

2 **Train value functions:**;

3 Train $V_{\text{Exploit}}$ on $\mathcal{D}_{\text{high}}$ using Eq. 1;

4 Train $V_{\text{Explore}}$ on $\mathcal{D}_{\text{low}}$ using Eq. 1;

5 **Train novelty ratio network:**;

6 Train $R_\psi$ using Eq. 15;

7 **for** *each epoch* **do**

8     **Train high-level policy $\pi_h$:** Update $\pi_h$ on $\mathcal{D}_{\text{high}}$ using Eq. 2

9     **if** *in warm-up phase (novelty ratio unstable)* **then**

10         Set $V_{\text{mix}}(s, g) \leftarrow V_{\text{Exploit}}(s, g)$

11     **else**

12         Set

$$V_{\text{mix}}(s, g) = (1 - R_\psi(s)) \cdot V_{\text{Exploit}}(s, g) + R_\psi(s) \cdot V_{\text{Explore}}(s, g)$$

13     **Train low-level policy $\pi_\ell$:** Sample mini-batch $(s_t, a_t, s_{t+1}, s_{t+k}) \sim \mathcal{D}_{\text{high}}$ Compute advantage

$$\tilde{A}_\ell(s_t, a_t, s_{t+k}) \approx V_{\text{mix}}(s_{t+1}, s_{t+k}) - V_{\text{mix}}(s_t, s_{t+k})$$

    Update $\pi_\ell$ using Eq. 3 with $\tilde{A}_\ell$

---

### A.4 LIMITATIONS AND DISCUSSIONS

In this paper, we have only done experiments based on noisy policy based suboptimal datasets. However, there are many other types of suboptimal datasets in addition to this, including vision occlusions, incomplete trajectories and so on. In different kinds of suboptimal datasets, the value functions and datasets structures may have different characteristics. We leave them as future works. What's more, the current algorithms are based on linear combinations of weighted value functions. However, we can also improve it as non-linear linear combinations of weighted value functions, which may be more accurate.

### A.5 USAGE OF LARGE LANGUAGE MODELS

We have used the Large Language Models to polish the writings and grammars. We claim that there are no other usage of the Large Language Models in addition to this.

