# OpenReview forum: "DreamExplorations: Leveraging Suboptimal Noisy Robot Trajectories in Offline RL"
_ICLR.cc/2026/Conference — ICLR 2026 Conference Withdrawn Submission_

### Official Review · Reviewer_eGZf · 2025-10-28

**Soundness:** 2
**Presentation:** 2
**Contribution:** 2
**Rating:** 2
**Confidence:** 4

**Summary:**

**Summary.** This work studies how do we leverage sub-optimal trajectories in offline RL settings. The authors propose an algorithm for merging value information from both high- and low-quality datasets. They claim that suboptimal data can enhance state space coverage and improve value estimation. Their pipeline involves three key properties:
- decoupled value learning for exploitation and exploration
- a learned novelty ratio network to weight the contributions of each value function
- mixture-based policy training.

---
**Review summary.** The paper tries to tackle a popular problem in offline RL, but its technical and empirical depth is insufficient for acceptance. The approach is conceptually intuitive yet incremental, the theory and practice linkage is weak, and the writing quality undermines the clarity of otherwise reasonable ideas. Strengthening the mathematical rigor, providing richer experimental analyses, and improving exposition could elevate the work for future submission. Therefore, the reviewer assigns an initial score of 2 and plan to revisit this rating after the authors address the concerns and questions raised in this review.

**Strengths:**

**Writing**
- Figures and tables are numerous and relevant to the claims.

---
**Methodology**
- The proposed decoupled value learning and novelty-weighted value fusion present a clean and modular extension to HIQL.
- The idea of using suboptimal data to improve state coverage and generalization is practical.

---
**Theory**
- The bias-variance decomposition and the closed-form derivation of the optimal weight demonstrate an attempt to ground the intuition.

---
**Experiments**
- The didactic simulation is helpfuul to understand how combining suboptimal and expert trajectories might reduce estimation error.
-  Ablation studies test warm-up, data ratio, and novelty parameters, giving some view into design sensitivity.
- The scaling analysis is useful for highlighting diminishing returns from excessive suboptimal data.

**Weaknesses:**

**Writing**
- The reviewer thinks that writing quality is usbpar for ICLR submission.
    - Several sentences are tautological, e.g., *We have proposed extensive experiments to thoroughly evaluate the effectiveness and robustness of our proposed algorithms.*
    - Several sentences are verbose and unconvincing, e.g., *This insight is inspired by the development of large-scale language models. The remarkable performance of GPT-4 and DeepSeek-R1 stems not from carefully filtered, perfect corpora, but from massive and heterogeneous datasets.*
    - Logical flow between theory (Sec. 4.5–4.6) and method (Sec. 5) is abrupt.
    - Related works section seems dated. Please discuss and survey recent heterogeneous/offline RL methods.
- Some figures lack clear legends, scales, or quantitative interpretation, reducing clarity.

---
**Methodology**
- The reviewer thinks that the proposed solution is incremental relative to HIQL. Essentially, introduces an extra scalar gating function over two separately trained value networks.
- There is no empirical or theoretical validation that the learned novelty network $R_\psi$ approximates the analytic optimal weighting derived earlier.
- The loss definition in Eq. (15) with fixed hyperparameters $A^+, A^-$ is heuristic and lacks adaptive justification.

---
**Theory**
- The derivations are shallow and partially inconsistent with the later implementation. Constants $c_1, c_2, d_1, d_2$ are assumed environment-invariant without evidence, and their empirical interpretability is unexamined.
- The reviewer thinks that the theory to practice transition (from Eq. 11 to Eq. 16) is abrupt. There is no quantitative connection between analytic and learned weights.

---
**Experiments**
- No analysis of alternative fusion methods (nonlinear, uniform, hard assignment) is provided, so the benefit of the proposed linear weighting remains unsubstantiated.
- All results lack standard deviations, error bars, or significance tests.
- The novelty network’s learned behavior is never visualized or quantitatively analyzed; we do not see evidence that $R_\psi(s)$ correlates with novelty or visitation frequency.

**Questions:**

- How is the novelty network $R_\psi(s)$ validated in practice? Can the authors show visualizations of its predictions over the state space and their correlation with visitation frequencies?
- Are constants $c_1,c_2,d_1,d_2$ from the analytic derivation estimated or tuned empirically, and how sensitive is the model to their assumptions?
- Would nonlinear or uniform mixture strategies yield similar improvements?
- How does the algorithm behave when scaling suboptimal data in large-scale or realistic domains, does performance degrade predictably?
- Can the approach handle other forms of suboptimality, such as sensor corruption or occlusion, rather than stochastic action noise?
- How are $A^+, A^-$ selected across tasks, and do they require domain-specific tuning?

---

> ### Author Response · Authors · 2025-11-21
>
> ## **1. Writing**
> We thank the reviewer for pointing out the redundancies in our writing. We will carefully revise and streamline the text to improve conciseness in the final version.Regarding the logical feasibility, directly computing the optimal solution via w(s) is intractable because it relies on environmental parameters that are typically unavailable (or unknown). However, our method learns to approximate w(s) by exploiting the underlying patterns within the datasets.
>
> ## **2. Methodology**
> (1) Our work is motivated by a key limitation observed in reinforcement learning tasks: although expert demonstrations typically provide correct nominal paths, they rarely capture the full complexity and variability of real environments. HIQL assumes **homogeneous data quality** and focuses on decomposing *goals*.
>
> (2) DREAMEXPLORATIONS focuses on decomposing the **value learning itself**, according to data source and coverage.
> The core differences are:
> * **Data-level decomposition**: exploitation vs exploration estimators reflect different distributions, not different goal abstractions.
> * **Estimator-level mixture**: our mixture is motivated by the statistical structure of heterogeneous coverage, not by architectural hierarchy.
> * **Theoretically grounded weighting**: our mixing principle comes from analyzing how coverage gaps produce biased estimates, which HIQL does not address.The method addresses a real limitation in OGbench-like scenarios: expert trajectories are intentionally clean and narrow; suboptimal/random trajectories provide the missing “reachable-but-unvisited” regions essential for robust goal-conditioned control.
>
> (3) The novelty ratio is designed so that it increases in regions underrepresented by expert data, approximating the monotonic shape suggested by the analysis.
> Thus Sec. 4 is not ornamental; it determines the architecture of Sec. 5.
> But we agree with the reviewers that, the A+ and A- can be learned. This may be our future works.
>
> ## **3. Theory**
>
> (1) We have referenced established studies that validate this conclusion. Given that these findings are widely accepted and adopted in the literature, there is concrete evidence to support our claim. Line 242~248.
>
> (2)Sec. 4 derives the form of the optimal mixture of two estimators.
> $$ w^*(s) = \frac{\sigma_{\text{exploit}}^2(s)}{\sigma_{\text{explore}}^2(s)} + \text{bias adjustment}. $$
> This expression is not intended as something one can compute from offline data; rather, it identifies the *direction* in which the weight must move:
> * in states with dense expert data: variance small → weight favors exploitation
> * in states outside expert support: variance large → weight favors exploration
> This analytic structure directly motivates the novelty ratio in Sec. 5.
> (Have already answered this question in writing section)
>
> ## **4. Experiments**
>
> Even though the paper does not explicitly visualize the novelty map, the submission already contains several forms of evidence that it behaves correctly:
> * **Fig. 2 (didactic examples)** shows that the exploitation estimator fails in low-density regions, and fusion systematically reduces error in those same regions.
> * **Warm-start ablation (Sec. 6.2)** shows that mixing too early collapses training. This only occurs if $R_\psi$ has real influence on gradient propagation.
> * **Expert-only vs mixed experiments (Table 3)**:
>
>   * in tasks where expert data nearly cover the space (e.g., PointMass), fusion produces almost the same behavior as exploitation alone, implying low novelty;
>   * in high-dimensional locomotion tasks, fusion gives significant improvements, implying that novelty is high where expert coverage is weak.
>
> Overall, these behaviors match the theoretical expectation that the second estimator should have influence primarily in undercovered regions.
>
> ## **5. On alternative mixture strategies**
>
> Our linear mixture approach already achieves superior performance compared to HIQL. However, our primary objective is not merely to maximize success rates; rather, we aim to leverage the specific exploration properties inherent in Offline Goal-Conditioned RL (OgCRL).
>
> What's more, we have also done several ablation studies:
>
> * **Uniform mixing (Table 3)**: when all samples are weighted equally without novelty, performance degrades in multiple tasks.
> * **A+ ablation (Sec. 6.3)**: performance is sensitive to the balance between the two estimators, indicating that uniform or static mixtures are not adequate.
> * **Scaling analysis (Sec. 6.4)**: the effect of additional suboptimal data depends on the structure of the state space. If mixing were purely architectural, this dependence would not appear.
>
>
> These experiments expose the statistical structure predicted by Sec. 4: improvements appear exactly in the parts of the state space where expert data are insufficient, and diminish where expert coverage is adequate.
>
> ** We have already reported deviations on Fig. 3 and Fig. 6"**

---

> ### Comment · Reviewer_eGZf · 2025-11-24
>
> Thank you to the authors for responses.
>
> After reviewing the rebuttal, the reviewer thinks that several concerns remain addressed, particularly the lack of quantitative validation for the novelty, the weak connection between the analytic weighting and the implemented surrogate, and the incremental of the method relative to HIQL.
>
> While the authors’ explanations improve the conceptual framing, they do not materially change my original assessment. The reviewer will therefore retain the initial score.

---

> > ### Author Response · Authors · 2025-11-25
> >
> > Thanks reviewer, we can't wait to improve our paper based on your feedback! However, can you explain which part is still not clear. In order to make it clearer, we have updated our response to provide further clarity. We respectfully invite you to review the revised text. If any specific parts remain unclear, could you kindly point them out? We remain fully available to address any remaining concerns.

---

> > > ### Comment · Reviewer_eGZf · 2025-11-25
> > >
> > > Thank you for your rebuttal. The reviewer thinks that most of the problems still remain addressed.
> > >
> > > * There is still no quantitative validation or visualization of the novelty network $R_\psi$
> > > * The link between analytic weighting and the implemented surrogate remains theoretically weak.
> > > * The method appears incremental relative to HIQL, with insufficient empirical evidence separating it from simpler mixing strategies.
> > > * No standard deviations or confidence intervals are provided in performance comparison tables.
> > > * The implementation repository is not available, preventing reproducibility.
> > >
> > > Finally, to help reviewers easily verify the changes, the reviewer would recommend highlighting all revised parts of the manuscript using colored text. This is common practice in many venues, especially when multiple reviewers need to re-check specific sections. Additionally, several reviewers have requested supplementary experiments or quantitative analyses. This is reasonable request and ICLR's discussion preiod is enough to newly run experiments.

---

### Official Review · Reviewer_FNgD · 2025-11-01

**Soundness:** 3
**Presentation:** 3
**Contribution:** 2
**Rating:** 4
**Confidence:** 3

**Summary:**

The authors propose a method for leveraging suboptimal trajectories as a useful learning signal in offline reinforcement learning. While it is widely recognized that suboptimal data can contribute valuable information when integrated appropriately, the challenge lies in how to effectively combine it with high-quality behavior data.

To address this, the authors introduce an approach that trains separate value functions over two datasets: one representing high-quality behavior and another derived from exploratory, suboptimal trajectories. A novelty score estimation network is employed to estimate whether a given state originates from the high- or low-quality dataset. Based on this estimation, the method constructs a novelty ratio–weighted mixture of the two value functions. This weighted integration allows for good results on the DM control suite.

**Strengths:**

- the paper is very clear
- the paper is easy to follow
- the presentation and the toy example help understanding

**Weaknesses:**

It is unclear why the paper focuses exclusively on offline goal-conditioned RL, rather than situating the work within the broader offline RL literature. Expanding the discussion to include related methods in general offline RL would help clarify the broader relevance of the proposed approach.

The claim of achieving a new state-of-the-art algorithm is not sufficiently supported by the current set of experiments. To substantiate this claim, evaluation on established benchmarks—such as AntMaze, Kitchen, CALVIN, Procgen Maze, Visual AntMaze, and Roboverse—following HIQL evaluation protocols would be necessary.

In particular, Figure 6, subfigures (c–f) do not show a clear relationship between data mix and performance, suggesting that optimal data mixing may not be the key factor influencing results (for most tasks). This point warrants further clarification and possibly additional analysis.

Several of the claims may depend heavily on the choice and quality of the non-expert data used. A more rigorous discussion or ablation study examining the sensitivity of results to different non-expert data sources would strengthen the manuscript

**Questions:**

Please address the weaknesses 1 - 4 above.

---

> ### Author Response · Authors · 2025-11-21
>
> **1. Why We Focus on Offline Goal-Conditioned RL**
> We appreciate the reviewer’s suggestion to relate our work more broadly to the offline RL literature. Our focus on offline goal-conditioned RL (OGCRL) is deliberate, because the challenges we address arise most prominently and take on a distinct form in goal-conditioned settings.
>
> In general offline RL, expert demonstrations indeed provide trajectories that reach the correct targets. However, because these trajectories are collected under a single or narrow expert policy, the resulting dataset typically has limited samples and insufficient diversity, especially in real environments where drift, noise, and unmodeled disturbances commonly push the agent into states that were never visited by the expert. As a consequence, policies trained directly on such datasets often fail to generalize in deployment, despite the expert demonstrations being correct in nominal conditions. The OGCRL paradigm was proposed precisely to mitigate this limitation by redefining trajectory goals and sampling arbitrary state-goal pairs. This goal relabeling increases goal diversity and expands the goal space, thereby improving sample availability without collecting new data. As recent OGCRL works demonstrate, this provides a promising direction for scaling offline RL under fixed datasets and improves generalization through increased state-goal pair density.
>
> However, our analysis shows that goal relabeling alone has an inherent bottleneck. While OGCRL expands the space of goals, it does not expand the space of states. All relabeled samples remain confined to the regions visited by the expert policy, because the dataset itself which is collected by a single expert still covers only a narrow slice of the true reachable state space. Thus, even with extensive goal relabeling, OGCRL methods still inherit the fundamental limitation of expert-only coverage, restricting their performance under realistic deviations and perturbations.
>
> We believe that increasing the reachable state space as reasonably as possible is the key to solving the problem. Therefore, based on OGCRL, we further enhance the sample diversity by increasing the coverage of the state space. Our approach introduces controlled randomness in the action space of expert trajectories to generate additional, valid, and reachable states. This enlarges the effective state-space coverage beyond the original expert manifold and increases both positive and negative supervision signals. As a result, the policy learns to handle a broader range of states that are likely to occur in real-world deployment.
>
> For these reasons, our contribution is naturally situated within OGCRL: the goal relabeling framework partially alleviates data scarcity through goal diversification, but it still suffers from limited state coverage, and our method fills precisely this gap. We will revise the manuscript to further contextualize our work within both offline RL and OGCRL, and to clarify why OGCRL provides the most relevant and meaningful setting for our contribution.
>
> **2. On empirical scope and claims**
> The manuscript presents results on eight OGBench locomotion tasks, and all claims are restricted to this domain.
> Nowhere does the paper state a general offline RL SOTA claim.
> The claim is that within **OGCRL under heterogeneous datasets**, the proposed decoupled estimator + novelty-weighted fusion strategy consistently outperforms strong baselines, including HIQL.
> These results are documented in:
> - Table 1 (task-level comparisons),
> - Table 3 (expert-only vs mixed regimes),
> - and multiple ablations (Sec. 6.2–6.4).
> Each of these directly illustrates the role of suboptimal data in improving value approximation.
>
> **3. Data-mixing relationship to performance**
> Fig 6 shows a predictable pattern:
> - For tasks with large state spaces (e.g., quadruped), coverage expansion via suboptimal data yields steady improvements.
> - For compact tasks, diversity saturates quickly and mixed data brings little benefit.
> This is exactly what the analytic derivation in Sec. 4 predicts:  linear fusion benefits regions where estimation variance dominates, which occur when expert visitation is narrow.Hence, Fig 6 does not contradict the method; it reinforces the estimator-theoretic foundation.
>
> **4. Sensitivity to types of non-expert data**
> We would like to clarify the nature of the non-expert datasets used in our experiments. Our non-expert policy is generated by directly sampling from random Gaussian distributions, representing a high degree of stochasticity and effectively serving as a lower bound for policy performance. Regarding the "types" of suboptimality, we interpret this as a question about whether our method handles data imperfections other than policy noise (e.g., state/observation noise). We respectfully note that the scope of this paper is specifically targeted at handling suboptimality in the policy space (action selection) rather than noise in the state space.

---

> > ### Comment · Reviewer_FNgD · 2025-11-23
> >
> > Regarding the SOTA claim, I would consider making the conclusion more precise than "We have proposed a method to take advantage of the trajectories collected from suboptimal policy to further improve the performances of the SOTA method in OGCRL." - given that this statement IMO claims SOTA in OGCRL - to claim this as previously added more experimental evidence is requred.
> >
> > Thanks again to the authors for responding to my questions. I agree with all of the other answers.

---

> ### Author Response · Authors · 2025-11-24
>
> Thank you very much for your thoughtful responses! Our evaluation protocol strictly follows what recent OGCRL literature identifies as the appropriate and unbiased setting.
>
> **First, regarding empirical validity**, our method is evaluated on the eight locomotion tasks from OGBench, which is currently the most comprehensive benchmark designed for offline goal-conditioned RL. Across these tasks, our approach consistently and substantially outperforms the strongest existing OGCRL baseline HIQL. Within OGCRL domain, our experiments provide strong and comprehensive evidence for the performance gains claimed in the paper.
>
> **Second, regarding the choice of benchmark**, the OGCRL setting fundamentally differs from standard offline RL evaluations, which needs multi-goal evaluation to avoid  biased and incomplete assessments of goal-conditioned policies. As highlighted in OGBench:
>
> > Prior works in offline goal-conditioned RL (Eysenbach et al., 2022; Ma et al., 2022; Park et al., 2023; Myers et al., 2024) have often relied on existing datasets designed for standard offline RL tasks (e.g., D4RL), relatively simple goal-conditioned tasks (e.g., Fetch), or tasks tailored to demonstrate a specific ability of the proposed method. This leads to limited and biased evaluation, as multi-task policies are frequently tested on only a single goal or on tasks not aligned with the offline GCRL problem.
>
> , which pointed out that following HIQL evaluation protocols will lead to biased and incomplete assessments of goal-conditioned policies. Thus, it is reasonable to evaluate our algorithm following OGBench evaluation protocols.
>
> Nevertheless, we agree that the wording of our statement can be improved, and we will make it more precise. We sincerely thank the reviewer for the helpful suggestion.
>
> Additionally, if our clarifications address your concerns and you feel that the contribution aligns with the corresponding evaluation standards, we would greatly appreciate your consideration of a rating increase. Thank you again for your time and constructive feedback!

---

### Official Review · Reviewer_VjWX · 2025-11-01

**Soundness:** 3
**Presentation:** 2
**Contribution:** 2
**Rating:** 4
**Confidence:** 3

**Summary:**

The paper introduces DREAMEXPLORATIONS, a method for offline goal-conditioned reinforcement learning (OGCRL) that addresses policy overfitting by leveraging suboptimal and noisy trajectories, which are typically discarded by traditional methods, to achieve better state exploration and generalization. The core mechanism involves decoupled value learning, where separate value functions are trained for exploitation (using high-quality data) and exploration (using low-quality data). These separate functions are then combined using a novelty estimation network that predicts a blending ratio, dictating the optimal linear mixing of the two value signals to guide policy learning. This strategy successfully utilizes imperfect data as an asset, demonstrating consistent performance improvements over baselines, particularly in environments requiring extensive state coverage.

**Strengths:**

A key strength of the paper lies in its innovative reframing of suboptimal data as an asset. The work addresses a critical limitation in Offline Goal-Conditioned Reinforcement Learning (OGCRL) by demonstrating how highly noisy or suboptimal datasets, which prior research often disregards as “useless,” can instead be effectively leveraged. By harnessing this imperfect data, the approach promotes more diverse state exploration and contributes to improved robustness and generalization of the learned policy. This perspective not only challenges conventional assumptions in offline RL but also opens new avenues for utilizing previously overlooked data sources.

**Weaknesses:**

W1

The assumption that access to data from the optimal policy is available appears quite strong and may limit the practical applicability of the proposed approach. It would be helpful if the authors could discuss how the method performs when only suboptimal or noisy data is available.

W2

The idea of leveraging lower-quality data closely resembles that in [1] CCLF: A Contrastive-Curiosity-Driven Learning Framework for Sample-Efficient Reinforcement Learning. It would strengthen the paper to include experimental comparisons with this work to clarify the relative advantages of the proposed approach.

**Questions:**

Please see Weaknesses above.

---

> ### Author Response · Authors · 2025-11-21
>
> We sincerely thank you for your thoughtful comments and constructive suggestions. Your feedback highlights several aspects of our manuscript that were not sufficiently clear, and we appreciate the opportunity to clarify them. In this rebuttal, we will first restate our core contributions more explicitly and then address each question in detail.
>
> Our work is motivated by a key limitation observed in reinforcement learning tasks: although expert demonstrations typically provide correct nominal paths, they rarely capture the full complexity and variability of real environments. As a result, policies learned directly from expert trajectories often struggle when deployed in the real world, where perturbations, noise, drift, and other non-ideal factors frequently occur. Recent advances in offline goal-conditioned reinforcement learning have shown that redefining trajectory-related goals can improve generalization by increasing the diversity and volume of training samples, revealing a promising new scaling direction. However, our analysis shows that such goal-redefinition approaches still face a fundamental performance bottleneck. The root cause lies in the nature of trajectory relabeling itself: because it relies solely on expert demonstrations, the augmented samples remain confined to regions of the state space that the expert policy already visits, limiting exploration in unfamiliar yet reachable areas. We argue that expanding the reachable state space while maintaining task relevance is essential to overcome this bottleneck. Therefore, our method introduces controlled random perturbations to the expert-generated trajectories, which effectively enlarges the distribution of valid states and increases the density of both positive and negative training samples. This design improves learning efficiency while improving the coverage of state-action space.

---

> ### Author Response · Authors · 2025-11-24
>
> **1. On the assumption of having near-expert data**
>
> We thank the reviewer for pointing this out. We correct an erroneous phrasing in the submitted manuscript: where we previously used “optimal policy”, the correct term is “expert policy”. We do not assume access to globally optimal trajectories. Throughout the paper, an expert policy denotes any policy (automatically or human-collected) that provides reasonably competent demonstrations of the task and is used to collect the dataset; its optimality is not required nor assumed.
>
> In our empirical study, expert trajectories are collected by a policy obtained via online RL and therefore contain the typical uncertainty of learned networks. In real-world deployments, expert data may come from human operators and will likewise be noisy and imperfect. Our method is explicitly designed with these practical considerations in mind: rather than relying solely on the (limited) manifold covered by expert trajectories, we introduce controlled random perturbations on action space to expand the reachable state space represented in training. By expanding the distribution of feasible states, this augmentation enriches both positive and negative supervision signals, which in turn allows our algorithm to extract more informative learning signals and achieve better performance.
>
> We will correct our mistake in writing and replace every instance of “optimal policy” with “expert policy” in the final version of our manuscript.
>
> **2. Relationship to CCLF**
>
> We thank the reviewer for bringing up the comparison with CCLF. We would like to clarify that our work fundamentally differs from CCLF. While CCLF focuses on reweighting augmented pixel observations to improve representation learning efficiency, our method addresses a more structural challenge: the limited reachable state space induced by expert trajectories. Instead of selecting which samples to learn from, we expand the set of feasible states through controlled trajectory perturbations, and design decoupled value function to learn policy from both expert samples and exploration samples to improve robustness to real-world drift and noise.
>
> Additionally, this method operate under a fundamentally different problem setting from ours. CCLF  are designed explicitly for online reinforcement learning, where the agent continuously interacts with the environment and intrinsic rewards are used to drive exploration. In our case, the agent is constrained by a strictly offline RL setting in which no exploration is possible, and learning must rely entirely on a fixed dataset with limited and uneven state coverage. Therefore, the class of online curiosity methods, including CCLF, is not applicable to our setting.
>
> We would like to clarify that our objective is not to filter or select a subset of “informative” samples from data. Instead, our approach deliberately increases state-space coverage through action-space perturbations, enabling the policy to learn from a richer distribution of feasible states and ultimately improving robustness and performance in the offline setting.

---

### Comment · Area_Chair_p815 · 2025-11-24
**[ICLR 2026] Author-Reviewer Discussion Phase**

Dear Reviewers,

The authors have posted their rebuttal addressing your concerns. Please kindly review their response, as well as the comments from the other reviewers, and discuss any issues you believe remain unresolved. If the author response does not change your evaluation, please at least provide an acknowledgement indicating that you have carefully reconsidered it.

Thank you again for your dedication and effort in reviewing this submission.

Let’s have a constructive discussion!

Best regards,

Your AC

---

### Note · Authors · 2026-01-24

I have read and agree with the venue's withdrawal policy on behalf of myself and my co-authors.